# PD-1 is imprinted on cytomegalovirus-specific CD4+ T cells and attenuates Th1 cytokine production whilst maintaining cytotoxicity

Helen M. Parry[1‡], Alexander C. Dowell[1‡], Jianmin Zuo[1], Kriti Verma[1], Francesca A. M. Kinsella[1], Jusnara Begum[1], Wayne Croft[1,2], Archana Sharma-Oates[2,3], Guy Pratt[1,4], Paul Moss[1]*

1 Institute of Immunology & Immunotherapy, College of Medical and Dental Sciences, University of Birmingham, Birmingham, United Kingdom, 2 Centre for Computational Biology, College of Medical and Dental Sciences, University of Birmingham, Birmingham, United Kingdom, 3 Institute of Cancer & Genomics, College of Medical and Dental Sciences, University of Birmingham, Birmingham, United Kingdom, 4 Queen Elizabeth Hospital, University Hospitals Birmingham NHS Foundation Trust, Birmingham, United Kingdom

‡ These authors are joint senior authors on this work.
* P.MOSS@bham.ac.uk

**Data Availability Statement:** Relevant data are within the manuscript and its Supporting Information files. All sequencing data files are

## Abstract

PD-1 is expressed on exhausted T cells in cancer patients but its physiological role remains uncertain. We determined the phenotype, function and transcriptional correlates of PD-1 expression on cytomegalovirus-specific CD4+ T cells during latent infection. PD-1 expression ranged from 10–85% and remained stable over time within individual donors. This 'set-point' was correlated with viral load at primary infection. PD-1+ CD4+ T cells display strong cytotoxic function but generate low levels of Th1 cytokines which is only partially reversed by PD-1 blockade. TCR clonotypes showed variable sharing between PD-1+ and PD-1- CMV-specific cells indicating that PD-1 status is defined either during T cell priming or subsequent clonal expansion. Physiological PD-1+ CD4+ T cells therefore display a unique 'high cytotoxicity-low cytokine' phenotype and may act to suppress viral reactivation whilst minimizing tissue inflammation. Improved understanding of the physiological role of PD-1 will help to delineate the mechanisms, and potential reversal, of PD-1+ CD4+ T cell exhaustion in patients with malignant disease.

## Author summary

PD-1 is expressed by a subset of CMV-specific CD4 T cells, we show expression is not associated with activation or 'exhaustion'. We go onto show the size of the PD-1+ pool is established at primary infection, and expression remains stable on antigen specific cell populations and on individual cells, indicating expression is imprinted and controlled by a 'set point'. PD-1 expressing CD4 T cells comprise cells with strong cytotoxicity but reduced cytokine production which may act to suppress viral reactivation whilst minimizing tissue inflammation.

available at https://www.ncbi.nlm.nih.gov/geo/query/acc.cgi?acc=GSE165952.

**Funding:** This research was funded by the Medical Research Council, https://mrc.ukri.org/, MR/R011230/1 to PM and by the Wellcome Trust, https://wellcome.ac.uk/, UNS29248 to HP. The funders had no role in study design, data collection and analysis, decision to publish, or preparation of the manuscript.

**Competing interests:** The authors have declared that no competing interests exist.

## Introduction

PD-1 is a member of the CD28 superfamily and is expressed widely on leucocyte populations [1]. Checkpoint proteins such as PD-1 and LAG-3 play an important physiological role in regulating the magnitude and function of the adaptive T cell immune response [2] and high levels of PD-1 expression are observed on exhausted T cells in the setting of cancer or chronic infection [3,4]. Moreover, therapeutic PD-1 blockade has demonstrated remarkable efficacy in the treatment of many patients with malignant disease [5]. However, despite the crucial therapeutic importance of PD-1 relatively little is known regarding its physiological role in the regulation of T cell function in healthy donors. This is particularly true for CD4+ T cells and is noteworthy given increasing appreciation of the role of CD4+ T cells in tumour-specific immunity [6].

Cytomegalovirus (CMV) is a highly prevalent human herpesvirus which establishes a state of persistent lifelong infection. Infection triggers a uniquely strong virus-specific T cell immune response which must be maintained in order to suppress viral reactivation [7]. CMV reactivation is an important cause of morbidity and mortality in patients who are immune suppressed, such as primary fetal infection or transplant recipients, but the CMV-specific immune response is highly regulated across the life course and infection is not thought to have major clinical consequence for healthy immunocompetent individuals.

CMV-specific CD4+ T cells play an essential role in control of viral reactivation [8], expand markedly within peripheral blood [9] and display a range of characteristic features including variable expression of CD27 and CD28 [10]. A unique feature of CMV-specific CD4+ T cells is their strong cytotoxic activity. Recognition of viral epitopes at the surface of virally-infected HLA class II+ target cells leads to target cell lysis and Th1 cytokine production. PD-1 is expressed on many CMV-specific CD4+ T cells [11] although the functional significance of this is unclear. Here we use HLA-class II tetramers to examine the phenotypic, functional and transcriptional correlates of PD-1 expression on CMV-specific CD4+ T cells. We show that PD-1 expression is present on a subset of virus-specific cells which is established at the time of primary infection and remains stable within individual donors. Furthermore, PD-1+ cells retain strong cytotoxic function but are markedly impaired in Th1 cytokine production. PD-1 + T cells therefore represent a discrete component of the physiological CD4+ T cell response to infection and understanding of their unique properties is likely to be crucial for uncovering mechanisms to reverse CD4+ T cell exhaustion in pathological states.

## Results

### The proportion of PD-1+ CMV-specific CD4+ T cells varies markedly within different individuals

HLA-Class II tetramers were used to identify CMV-specific CD4+ T cells within the blood of healthy CMV-seropositive donors. Tetramers were available against three immunodominant epitopes, two from pp65 and one from gB, although it must be noted that these only identify a proportion of the global CMV-specific CD4+ T cell response. The profile of PD-1 expression on tetramer-positive cells was then determined (Fig 1A), assessing both the proportion of cells that expressed PD-1 and the mean fluorescence intensity of staining. PD-1 was expressed on an average of 23% of CMV-specific CD4+ T cells with a range between 1.2% to 49% of the virus-specific population. Interestingly, this proportion was much greater than seen on the total CD4+ population (mean 11%; p = <0.0001). The MFI of PD-1 expression on the virus-specific pool also revealed ten-fold variation but was again much higher than on the global CD4+ population (MFI: 524 vs 297 respectively; p = <0.0001) (Fig 1B). A low level of non-specific tetramer binding was revealed through the use of control tetramers but had no influence

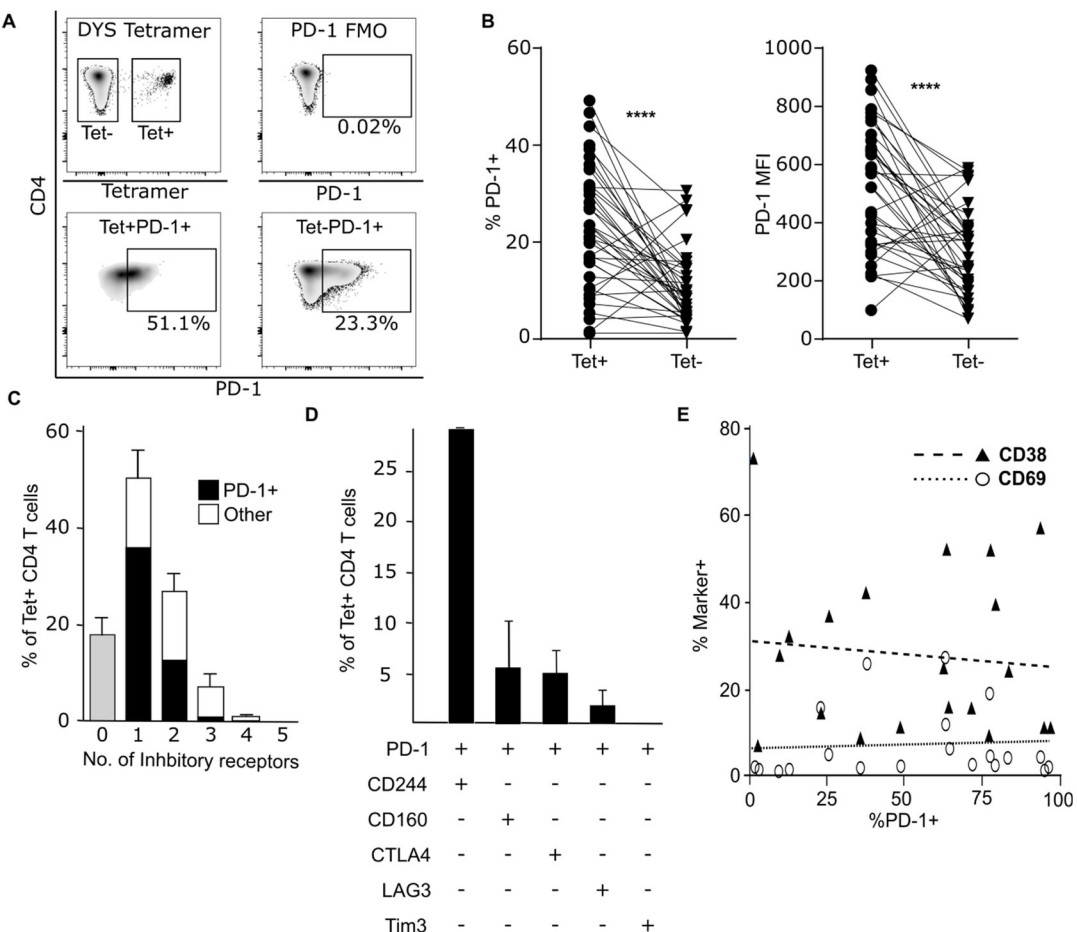

**Fig 1. CD4+ CMV-specific T cells express PD-1 but not in association with other inhibitory receptors or markers of activation.** (**A**) A representative example of PD-1 and tetramer staining. Expression of PD-1 on both CD4+T cells and the tetramer population is shown. (**B**) Combined data of PD-1 expression compared between CMV tetramer+ CD4+ T cells and the tetramer- CD4+ T cell population in healthy donors (n = 38), data show percentage positive and mean fluorescent intensity of whole populations. **** p = <0.0001. (**C**) The Proportion of tetramer+ CD4+ T cells expressing the inhibitory receptors (PD-1, CD244, CD160, CTLA4, LAG3 and Tim3) alone or in combination was determined by flow cytometry in healthy donors (n = 9), ■ indicates expression of PD-1 alone or in combination with other inhibitory, □ indicates expression of other inhibitory receptors without PD-1 expression. (**D**) Of cells expressing PD-1, the majority express only one other inhibitory receptor. Data shows expression of inhibitory receptors in addition to PD-1 by PD-1+ tetramer+ CD4 T cells expressing two inhibitory receptors. (**E**) The frequency of tetramer+ CD4+ T cells expressing PD-1 or either the early activation marker CD69 or late activation marker CD38 was determined in healthy donors (n = 20). No correlation was observed between the expression of PD-1 and CD69 ($R^2$ = 0.006) or CD38 ($R^2$ = 0.01). P values were calculated by two tailed paired t test, correlation was determined by linear regression. Error bars indicate SEM.

on the percentage expression of PD-1 (S1 Fig). The majority of CMV-specific CD4+ T cells had an effector phenotype (CCR7-CD45RA-) with increased CCR7-CD45RA+ Temra populations in some donors. PD-1 expression was not influenced by $T_E$ or $T_{EMRA}$ phenotype (S2 Fig).

A characteristic feature of T cell exhaustion during chronic infection is that PD-1 is typically expressed in association with additional checkpoint proteins [12]. As such we next determined the profile of PD-1 co-expression with the additional checkpoint proteins CTLA-4, LAG-3, CD244, CD160 and TIM3. Expression of at least one checkpoint protein was seen on 83% of tetramer positive cells and, of these, 60% of cells expressed PD-1 as the sole marker. 32% of cells expressed a second checkpoint protein whilst expression of three or more checkpoint proteins was seen in the remaining 8% of cells (Fig 1C). CD244 (2B4) was the most

common molecule expressed in combination with PD-1, followed by CD160, CTLA-4 and LAG-3, but no expression of Tim3 was observed (Figs 1D and S3).

PD-1 is expressed transiently following T cell activation and we therefore investigated the activation status of virus-specific CD4+ T cells in relation to PD-1 expression. Expression of the early activation marker CD69 was low on CMV-specific CD4+ T cells, with a median frequency of 4.2%, and no association was observed with PD-1 expression (Non-linear regression analysis: $R^2 = 0.005$; p = 0.85). CD38 is a marker of late activation and was present on 14% of the virus-specific pool, but again unrelated to PD-1 expression ($R^2 = 0.043$; p = 0.57) (Fig 1E). As such the expression of PD-1 on CMV-specific CD4+ T cells is not related to markers of activation or exhaustion.

## PD-1 expression is correlated with viral load at primary infection and remains stable within individuals

Given the wide variation in the proportion of PD-1 expression on CMV-specific CD4+ T cells between individual donors we next went on to assess the determinants of this value and its stability over time. The level of initial antigen exposure may influence the PD-1 setpoint but as primary CMV infection is almost always clinically silent this was not possible to determine within the donor panel. However, a primary CMV immune response may develop following hemopoietic stem cell transplantation if a CMV seronegative graft is administered into a CMV seropositive patient. As such, the level of peak viremia was determined in this setting and assessed in relation to the level of PD-1+ expression on CMV-specific CD4+ T cells taken 18–26 weeks after resolution of viremia. The proportion of PD-1+ cells was seen to be correlated strongly with viral load (p = 0.017, $R_2 = 0.7469$) (Fig 2A). No correlation was seen between peak viral load and total frequency of tetramer-positive cells or PD-1+ tetramer-negative cells (S4A–S4C Fig).

In order to assess the stability of PD-1 expression, we also analysed serial blood samples from 16 healthy donors at 8 timepoints over a 12 month period. Strikingly, the proportion of PD-1+ cells was very stable within individual donors, with no significant difference at any time across the 12 months. This confirms that each individual has a 'setpoint' of PD-1 expression that remains largely constant over time (Fig 2B).

The stable composition of the PD-1+ CMV specific CD4 T cell pool *in vivo* suggested that the pattern of PD-1 expression is largely fixed on individual cells. To further explore this PD-1 + and PD-1- CMV-specific CD4+ T cells were cloned by limited dilution and the relative stability of PD-1 expression examined during culture. PD-1+ and PD-1- clones retained the PD-1 state from which they were derived *ex vivo* (Fig 2C). PD-1 phenotype was not affected by reactivation *in vitro*, remaining stable over a 4-week incubation following initial stimulation (Fig 2D). It was noteworthy that PD-1+ T cells largely retained CD28 expression whereas a significantly lower proportion of PD-1- clones expressed CD28 (Fig 2E). Relative CD28 expression in relation to PD-1 was not correlated with initial peak viral load (S4D Fig).

## PD-1 expression does not impair cytotoxic function

We next went on to investigate transcription factor expression and relative functional activity of PD-1+ and PD-1- CMV-specific CD4+ T cells. Tetramer-positive cells expressed high levels of both T-bet and Eomes in comparison to the total CD4+ repertoire but no differences were seen in relation to PD-1 expression (p = 0.34) (Fig 3A and 3B). Expression was also comparable between PD-1+ and PD-1- T cell clones *in vitro* (Fig 3C).

CMV-specific CD4+ T cells typically have strong cytotoxic activity. Intracellular expression of perforin and granzyme was therefore assessed in relation to PD-1 status. The majority of virus-specific cells exhibited dual expression of perforin and granzyme and no differences

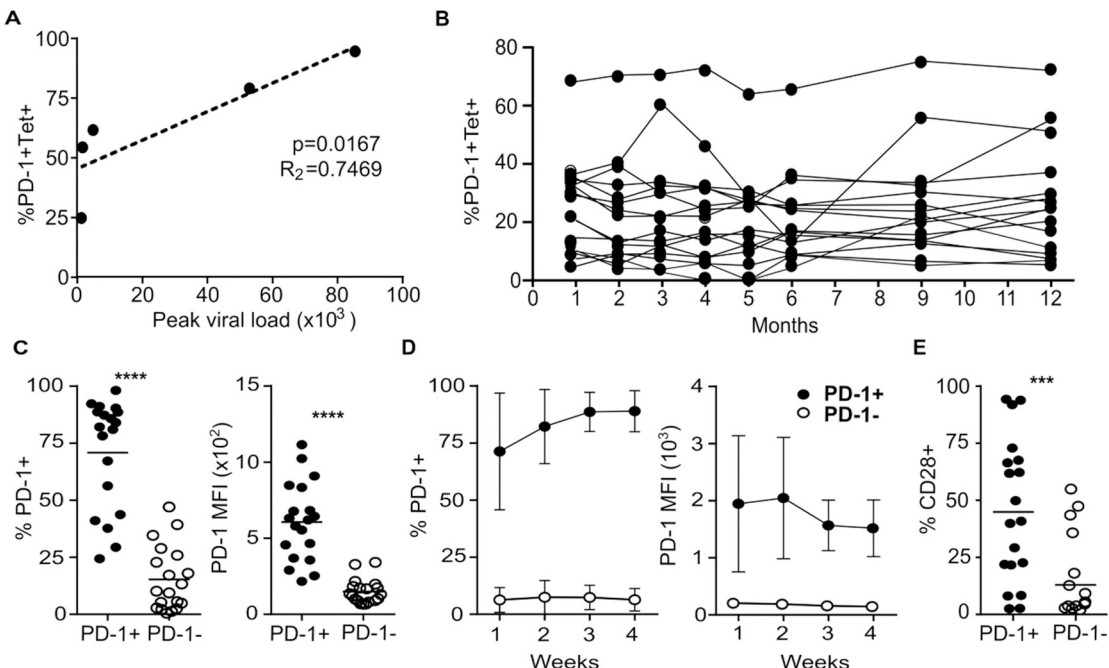

**Fig 2. The relative expression of PD-1 on CMV-specific CD4 T cells is associated with the peak viral load and is stable at a set-point through imprinting of individual T cells.** (**A**) Correlation of peak CMV viral load during acute infection in bone marrow transplant patients (n = 5) and the frequency of PD-1+ expression at 25 weeks post resolution of viremia. A positive correlation (p = 0.0167) was observed between the peak viral load and the frequency of PD-1 expression post resolution. (**B**) The frequency of PD-1 expression by CMV tetramer+ CD4+ T cells was monitored by flow cytometry over the course of 12 months in healthy donors (n = 16). Little variation was seen over the period or between the initial and final sample. (**C**) Following limited dilution cloning of cells derived from flow sorted PD-1+/- CMV tetramer+ CD4+ T cells from healthy donors (n = 3). The frequency of PD-1 expression was determined by flow cytometry on clones derived from either PD-1+ or PD-1- cells ex vivo (n = 38). **** p = >0.0001. (**D**) The expression of PD-1 was assessed weekly after subjecting the clones derived from PD-1+ or PD-1- T cells to stimulation. Clones remained stable for over 4 weeks in culture after stimulation (n = 20). (**E**) The frequency of CD28 expression is demonstrated on PD-1+ and PD-1- derived clones (n = 38). *** p = 0.0005. P values were calculated by two tailed t test, correlation was determined by linear regression. Error bars indicate SD.

were observed in relation to PD-1 expression (Fig 3D). Direct *ex vivo* cytotoxic function was determined by analysis of killing of peptide-pulsed target cells by selected PD-1+ and PD-1- antigen-specific T cells. In keeping with the shared profile of cytotoxic granule expression, both subsets demonstrated equivalent and efficient lysis of target cells (Fig 3E).

## PD-1 expression attenuates Th1 cytokine production and this is only partially reversible by PD-1 blockade

We next assessed the relationship between PD-1 expression on peptide-specific CD4+ T cells and their profile of cytokine production in response to cognate antigen stimulation. Lymphoblastoid cells (LCL) demonstrate high level expression of PD-L1 (28) and PBMC were therefore incubated with peptide-pulsed lymphoblastoid target cells. PBMC were then co-stained with CMV specific tetramer and the profile of cytokine production determined by intracellular flow cytometry. PD-1-negative T cells demonstrated a strong profile of Th1 cytokine production but this was attenuated substantially within PD-1 positive cells (Fig 4A and 4B). In particular, IFN-γ and TNF-α production was observed in 42% and 43% of PD-1 cells respectively, compared to only 11% and 7.4% of the PD-1+ pool (Fig 4B). IL-2 was not produced significantly by either PD-1+ or PD-1- CMV-specific cells (Fig 4C and 4D).

We next went on to examine the potential ability of antibody-mediated PD-1 blockade to enhance the profile of cytokine production by PD-1+ CMV-specific CD4+ T cells. PBMC were

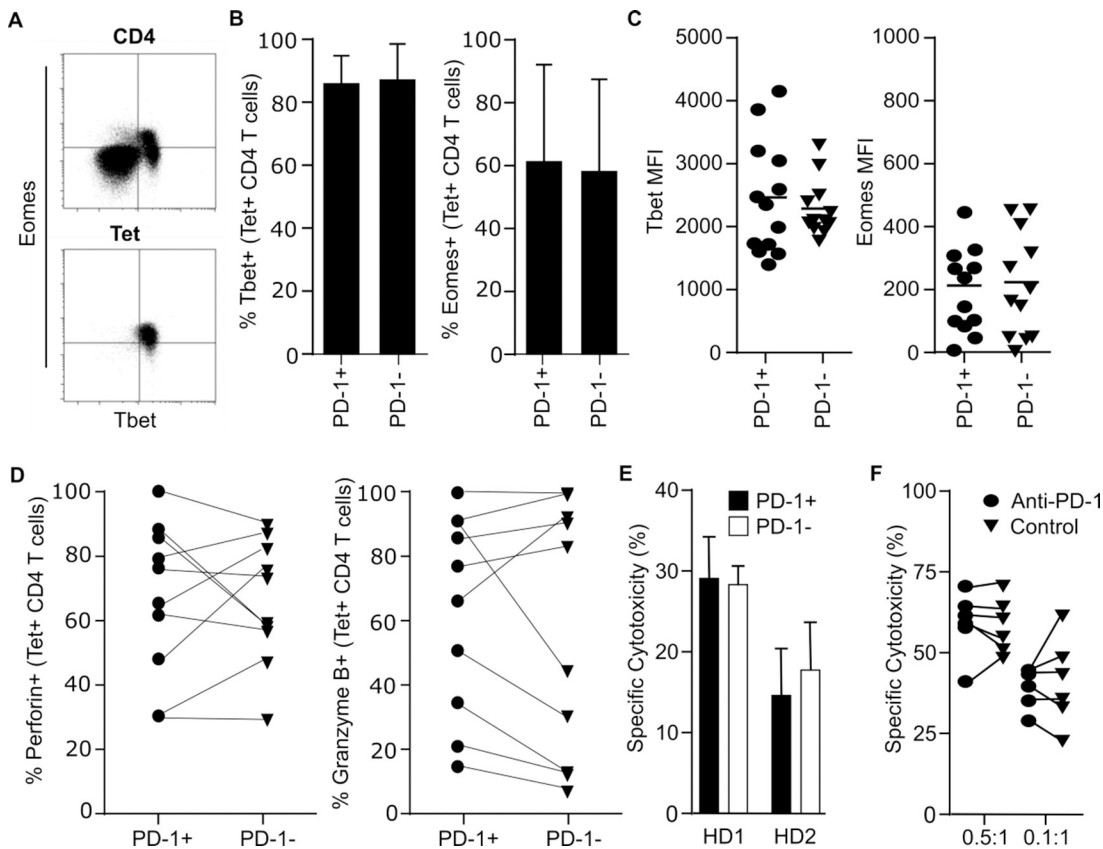

**Fig 3. The expression of PD-1 on CD4+ Tetramer+ cells does not impact on cytotoxicity.** (A) Example flow plots demonstrating the intracellular staining of CD4+ T cells and tetramer positive cells for the transcription factors Tbet and Eomes. (B) Combined data showing the frequency of Tbet and Eomes expression amongst PD-1+ and PD-1- CMV tetramer+ CD4+ T cells in healthy donors (n = 17). (C) The level of Tbet and Eomes staining amongst T cell clones derived from PD-1+ and PD-1- CD4+ T cells ex vivo (n = 25). (D) Expression of the cytotoxic granules perforin and granzyme B was determined by flow cytometry in healthy donors (n = 10) in PD-1+ and PD-1- CMV tetramer+ CD4+ T cells. (E) CMV Tetramer+ CD4+ T cells were flow sorted ex vivo and co-cultured with HLA-matched LCLs pulsed with relevant peptide. After 16hrs cytotoxicity was measured by absolute quantification of live LCLs by flow cytometry, in comparison to control cultures. Data is shown for two healthy donors in independent experiments. Error bars indicate SD (B), SEM (E). (F) Blockade of PD-1+ T cells with antibody against PD-1 does not suppress cytotoxic activity against peptide-pulsed target cells. Tetramer-positive cells were isolated directly from PBMC and incubated with target cells for 4 hours, in the presence (PD-1) or absence (Control) of inhibitory antibody against PD-1.

stimulated with CMV lysate, in the presence or absence of PD-1-specific antibody, and intracellular cytokine production was then determined. PD-1 blockade increased the profile of IFN-γ production by an average of 40% (2.7% vs 3.8%; p = 0.031) although this remained much lower than seen with PD-1- cells (Fig 4E and 4F). The response of individual donors to PD-1 blockade was variable but did not appear related to the presence of PD-1 ligands which was consistent across individuals (S5 Fig).

## TCR clonotypes of PD-1+ virus-specific T cells may be either unique or shared with PD-1- cells

The functional differences that were observed between PD-1- and PD-1+ T cells led us to consider the clonal relationship between these two subsets. In particular it was of interest to determine if PD-1+ populations represented unique clonal expansions or if an identical clonotype could simultaneously contain PD-1+ and PD-1- subpopulations. Single cell TCR sequencing

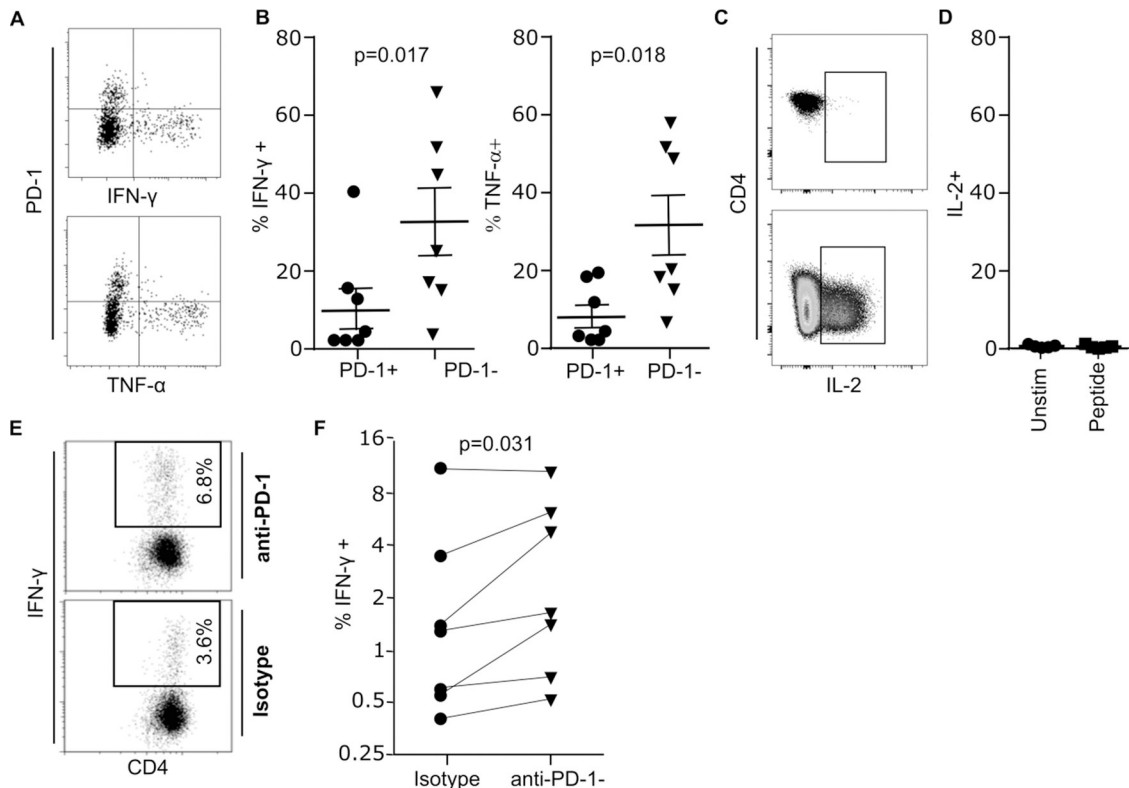

**Fig 4. PD-1 expression impairs Th-1 cytokine responses and can be partially reversed through PD-1 blockade.** Whole PBMCs were stimulated with HLA-matched, peptide pulsed LCLs and IFN-γ and TNF-α release was measured in CMV tetramer+ CD4+ T cells by intracellular staining. (**A**) An example showing CMV tetramer+ CD4+ T cell IFN-γ and TNF-α expression versus PD-1. (**B**) Pooled data of the frequency of cytokine expression in CMV tetramer+ CD4+ T cells in healthy donors (n = 7). (**C**) An example showing CMV tetramer+ CD4+ T cell (upper) IL-2 expression following peptide stimulation and IL-2 expression by total CD4+ T cells in response to PHA (lower). (**D**) Pooled data of the frequency of IL-2 expression in CMV tetramer+ CD4+ T cells in healthy donors (n = 5). (**E**) A representative flow plot of IFN-γ expression by intracellular staining of whole PBMC stimulated with CMV lysate, with PD-1 blocking or isotype control antibody. (**F**) Combined data of IFN-γ production as in (**E**) in healthy donors (n = 7). P values were calculated by two tailed paired t test (**B**) and Wilcoxon matched paired t-test (**F**).

of PD-1- and PD-1+ CMV-specific CD4 T cells was therefore undertaken on cells isolated directly *ex vivo* (S6 Fig). Analysis was performed on samples from three donors, two of which were specific for the DYS peptide and one with specificity for AGI peptide. PD-1+ and PD-1- clonotypes specific for DYS demonstrated a dominant and shared clonotype within each donor (Fig 5A and 5B) indicating that a single T cell clone can simultaneously comprise populations of PD-1+ and PD-1- cells. In contrast, the AGI-specific T cell response was largely polyclonal but a minority of clonotypes were also shared between PD-1+ and PD-1- subpopulations. Clonotype sharing was also reflected within T cell clones *in vitro* (S7 Fig) Thus, single T cell clonotypes can simultaneously comprise populations of PD-1+ and PD-1- cells.

## Differential gene expression is observed in PD-1+ and PD-1- CMV-specific CD4+ T cells

In order to investigate potential correlates of PD-1 expression we next undertook transcriptional analysis of PD-1+ and PD-1- CMV-specific CD4+ cells. Tetramer positive cells were sorted directly *ex vivo* prior to immediate RNA-Seq analysis (S4 Fig).

Differential expression was observed in 65 transcripts at p<0.01 and 15 remained significant at p<0.001 (Fig 6A and 6B). 11 genes were upregulated in PD-1+ cells including *PDCD1*

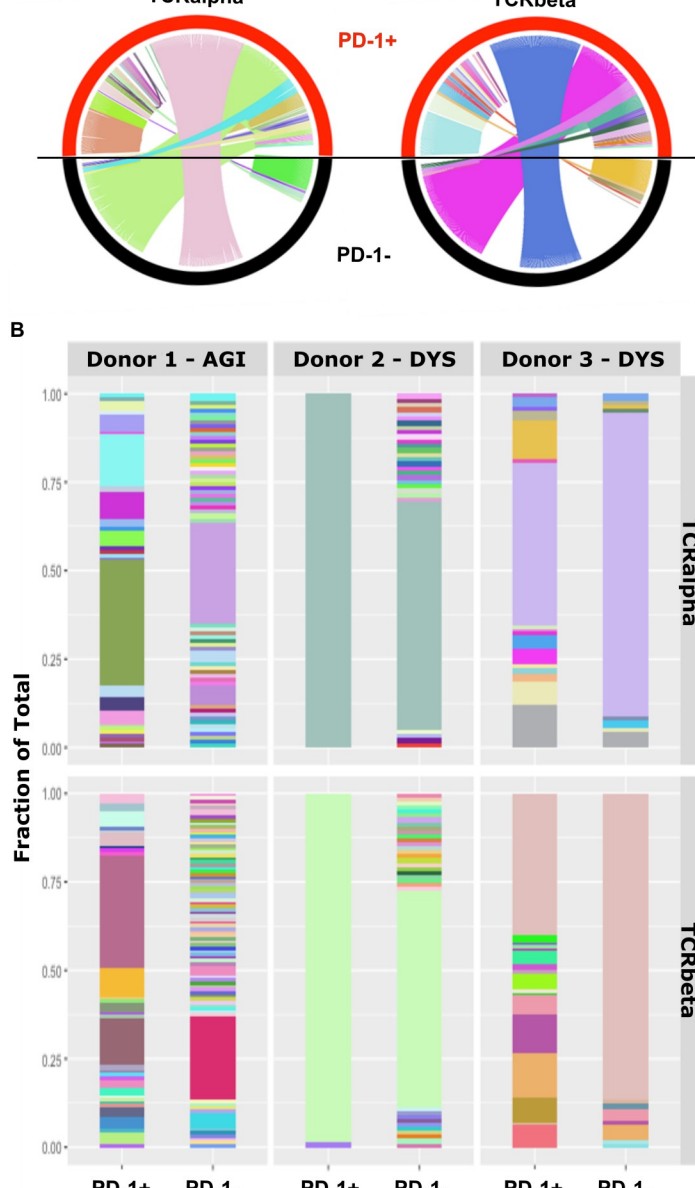

**Fig 5. TCR clonotypes are shared between PD-1+ and PD-1- CMV-specific T cells.** CMV Tetramer+ CD4+ T cells were ex vivo single cell sorted ex vivo dependent on PD-1 status. Results are shown for single cell sequencing of individual TCRalpha and TCRbeta CDR3 regions of individual cells. (**A**) Circos plots are shown demonstrating the relationship of TCRalpha and beta CDR3 sequences between PD-1+ and PD-1- CD4+ tetramer+ cells (n = 3 donors). (**B**) TCR sequence data presented showing individual donor repertoires, as a proportion of the total sequenced cell population.

(PD-1), granzyme K, CD28 and IL-12Rβ2. 4 genes were downregulated in PD-1+ cells including the zinc finger transcription factor ZNF683 (Hobit), a major regulator of IFN-γ production and cytotoxic activity [13], and protein expression was assessed by flow cytometry (S8A Fig). Gene Set Enrichment Analysis (GSEA) of signature gene sets was also performed (S8B Fig).

In order to assess if the transcriptome of PD-1+ CD4+ virus-specific cells was comparable to those previously analysed from 'exhausted' T cells we next compared our PD-1

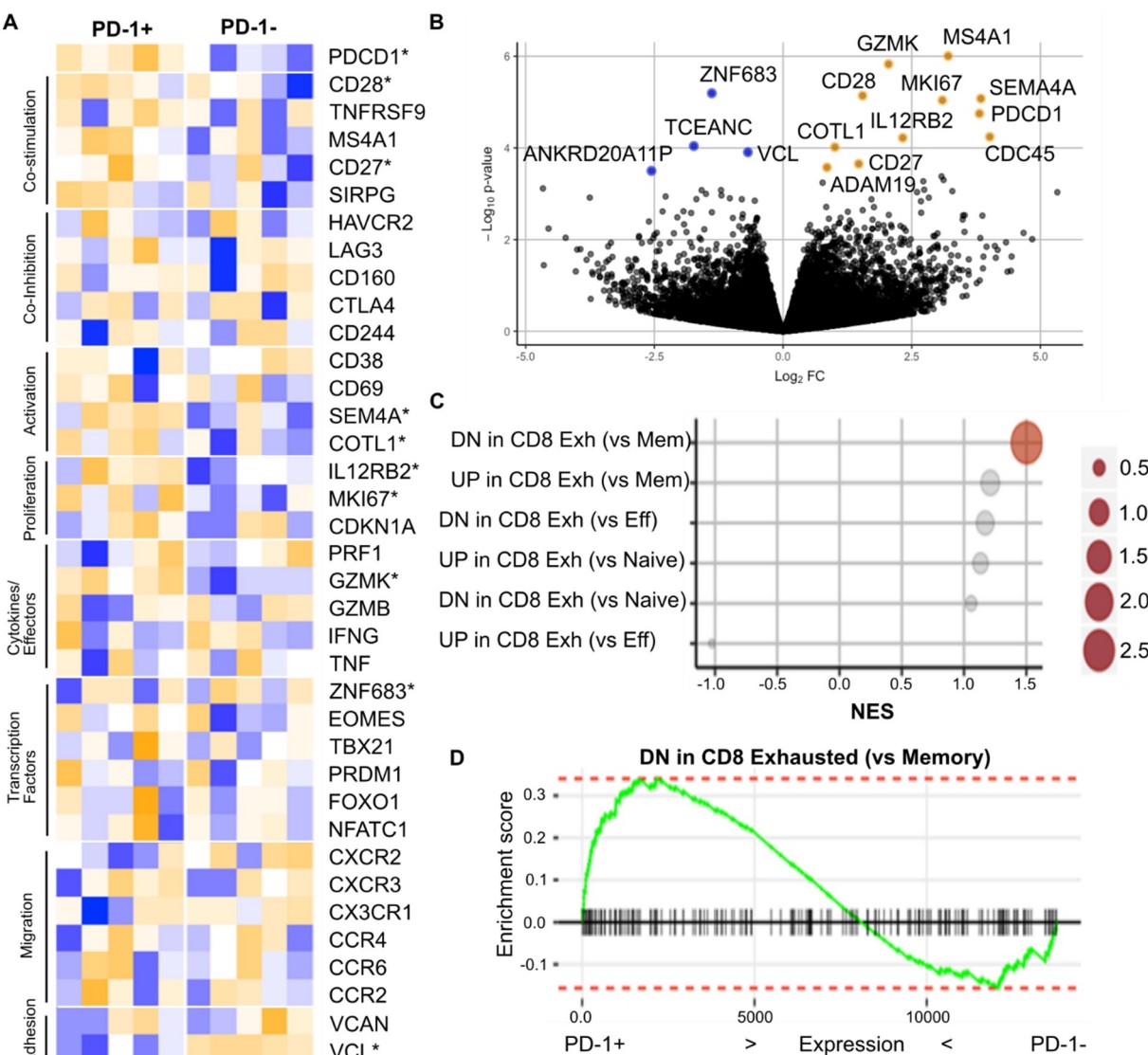

**Fig 6. RNA Sequencing of PD-1 positive and PD-1 negative tetramer+ CD4+ T cells.** (**A**) Transcriptional profile of selected genes of interest in PD1+ and PD1- CMV tetramer+ CD4+ T cells sorted ex vivo. Colour scale represents standard deviations from the mean; *Differentially expressed (p < 0.0001) gene from PD1+ vs PD1- differential expression analysis. (**B**) Differential expression analysis results from PD1+ vs PD1- CMV-specific CD4+ T cells. Genes labelled are those differentially upregulated (Orange) or downregulated (Blue) in PD1+ compared to PD1- cells. (**C**) Gene set enrichment analysis results for publicly available 'T cell exhaustion' signatures. Size of points represents–log10(FDR) and significant (FDR < 0.1) enrichment is highlighted in red. (**D**) Barcode plot of significantly enriched gene set from (c). This set consists of genes downregulated in exhausted CD8 T cells compared to memory cells. Vertical lines mark the position of these genes along the fold change ranking of PD-1+ vs PD-1- CMV tetramer+ CD4 T cells. Expression of genes downregulated in exhaustion are inversely associated with PD-1 + CMV tetramer+ CD4 T cells.

+ transcriptome with publically available datasets (Fig 6C and 6D). No such correlation was observed and, indeed, the one significant association demonstrated the opposite effect with increased expression of genes in PD-1+ cells that are downregulated in exhausted CD8 T cells compared to memory T cells. This is compatible with a regulatory role for PD-1 in limiting T cell differentiation during prolonged antigen exposure [14]. As such these data show there is no evidence from gene expression analysis or phenotypic data that PD-1+ CMV-specific CD4

+ T cells are exhausted. Rather these cells represent a specific subset of cells with unique functional capacity.

## Discussion

Therapeutic blockade of PD-1 engagement is used widely in management of malignant disease [5] but relatively little is known about the physiological role of PD-1 within human CD4+ T cell function. We studied PD-1 expression on CD4+ T cells specific for cytomegalovirus and as most people become infected with CMV during their lifetime the virus might potentially be considered as part of the human 'virome'. As such CMV represents a very different infectious challenge from agents such as HIV and hepatitis C which are relatively recent infections during the evolution of Homo sapiens and are poorly adapted to the human host; often leading to substantial pathology. PD-1 expression on HIV- or hepatitis C-specific T cells is considered as a marker of cellular exhaustion [15–17] and, indeed, antibody-mediated PD-1 blockade was initially developed for treatment of these conditions [12].

In contrast, CMV reactivation is effectively controlled within immunocompetent people and the CMV-specific immune response cannot therefore be considered as exhausted. As such, the findings reported here relate to the functional importance of PD-1 in regulation of physiological virus-specific CD4+ T-cell responses during health [18]. In this regard it should also be noted that the absolute level of PD-1 expression is lower on CMV-specific T cells in this setting than the PD-1hi phenotype of exhausted T cells [19]. A PD-1hi phenotype may, however, be seen in the setting of exhausted CMV-specific responses in highly immune suppressed transplant patients [20,21]. In addition, only a minority of PD-1+ CMV-specific cells expressed additional inhibitory co-receptor, a phenotype associated with exhaustion [22,23]. As such these cells are more similar to PD-1int populations that have been described in LCMV infection with stem cell and progenitor properties [24].

The proportion of PD-1+ virus-specific T cells was variable within different people and a striking finding was that these individual levels were stable for at least 12 months, indicating a 'setpoint' for PD-1 expression. Importantly, PD-1 expression status was retained on T cell clones following prolonged culture *in vitro* indicating epigenetic imprinting as a likely mechanism of regulation. It is interesting to speculate on the potential determinants that establish this setpoint. Murine studies indicate that the peak viral load at the time of primary infection determines PD-1 expression [23] and we also observed that the proportion of PD-1+ T cells within a tetramer-specific pool was correlated with the peak level of viremia at the time of primary infection. These data are from a limited number of patients, who are also immunosuppressed at time of infection, but may suggest that the size of the PD-1+ T cell pools expands in proportion to initial viremia whilst the PD-1- effector pool reflect a more stable baseline response following infection. Future studies will be required to further assess these findings following natural infection of healthy individuals.

Analysis of TCR clonotypes was undertaken in order to assess if PD-1+ cells represented a separate clonal lineage within a peptide-specific CMV-specific CD4+ response. Responses against the DYS peptide in two different donors showed that clonotypes were shared between PD-1- and PD-1+ populations, suggesting that PD-1 expression can be regulated after initial T cell priming and ruling out TCR affinity as a primary determinant of PD-1 expression. In contrast, there was less sharing of TCR clonotypes between PD-1+ and PD-1- CD4+ cells specific for the AGI peptide, revealing that PD-1 status may also become defined on different T cell clones at the time of initial T cell priming. The mechanisms determining PD-1 imprinting on individual T cells have not been fully defined at this time but the context of antigen encounter

is a significant determinant in an acute setting [25]. Epigenetic imprinting at the PD-1 locus is likely to define surface phenotype [26,27] and represents an important area for future study.

CMV-specific CD4+ T cells are highly cytotoxic and lyze HLA class II+ target cells such as dendritic cells and endothelial cells at the time of viral reactivation. These sites are major reservoirs for CMV infection and also express the PD-1 ligands PD-L1 and PD-L2. As, such PD-1 mediated regulation is likely to play an important role in fine tuning of CD4+-mediated viral control [28]. We found that PD-1+ cells retain potent cytotoxic activity but show reduced levels of Th1 cytokine production after antigen engagement. A key finding related to the relative functional activity of PD-1+ virus-specific T cells. A selective impact of PD-1 expression on cytokine production has been observed previously [23,29] although studies using CD8+ cells have indicated that cytotoxic function is also sensitive to PD-1 signalling [30]. As such, the physiological role of PD-1 on CD4+ T cells is to selectively limit cytokine release whilst maintaining cytotoxic function after antigen engagement. In this context it is known that CMV reactivation must be controlled very rapidly by target cell lysis in order to prevent viremia [11] and this function has clearly been retained by PD-1+ cells. In contrast, local release of Th1 cytokines at the site of viral activation may have the potential to mediate localised pathology, it is therefore possible that PD-1+ populations act to limit this complication and minimise the development of CMV-mediated pathology [31]. Here it is also important to consider that CMV undergoes latent infection within the endothelium which are subject to vascular surveillance by strong CMV-specific immune response within the blood [32]. PD-L1+ endothelial cells are critical in the maintenance of vascular integrity during chronic infection [33] and may represent a further rationale for the importance of PD-1 expression on CMV-specific T cells. Investigations utiltizing appropriate *in vitro* or *in vivo* models will be important to explore this aspect of constitutive PD-1 expression further.

A characteristic and relatively unique feature of CMV-specific CD4+ T cells is loss of CD28 expression on many cells during clonal maturation [10]. Of interest, PD-1+ CMV-specific CD4+ populations comprised both CD28+ and CD28- cells although most PD-1+ clones *in vitro* retained CD28 expression. This suggests that PD-1+ cells are relatively less differentiated than PD-1- subsets and is in keeping with the proposed role of CD28 in mediating the inhibitory activity of PD-1 engagement [34]. However, some PD-1+ T cell clones had lost CD28 expression and recent findings have shown that PD-1 functions as the rheostat of T cell activation rather than an inhibitor of a specific stimulatory co-receptor [35]. Relative retention of CD28 suggests that a further important role for PD-1 expression may relate to relative limitation in the degree of differentiation of CMV-specific T cells and prevention of terminal exhaustion [14]. In this regard it is noteworthy that primary infection with CMV is often acquired within the first year of life and as such infection must be controlled for over 70 years in many people [36].

Transcriptional analysis revealed that a relatively small number of genes were differentially expressed in PD-1+ or PD-1- virus-specific cells. PD-1+ cells showed strong downregulation of ZNF683 (Hobit), a major transcriptional regulator within the CD28- CMV-specific T cell pool [13]. Hobit positively regulates IFN-γ production and its downregulation in PD-1+ cells may thus partially explain the associated functional reduction in Th1 cytokine production [37]. However it is also associated with expression of perforin and granzyme and, as such, its downregulation within PD-1+ cytolytic CD4+ T cells is somewhat surprising. Potential confounding factors include the fact that Hobit is only expressed within the CD28- subset and is also downregulated after recent activation. Blimp-1 induces and Hobit maintains the cytotoxic mediator granzyme B in CD8+ T cells. Nevertheless, Hobit is closely related to Blimp-1 [38] and Blimp-1 has itself been shown to regulate PD-1 expression [39]. As such the potential importance of Hobit expression in relation to the PD-1 status of CD4+ cells deserves further

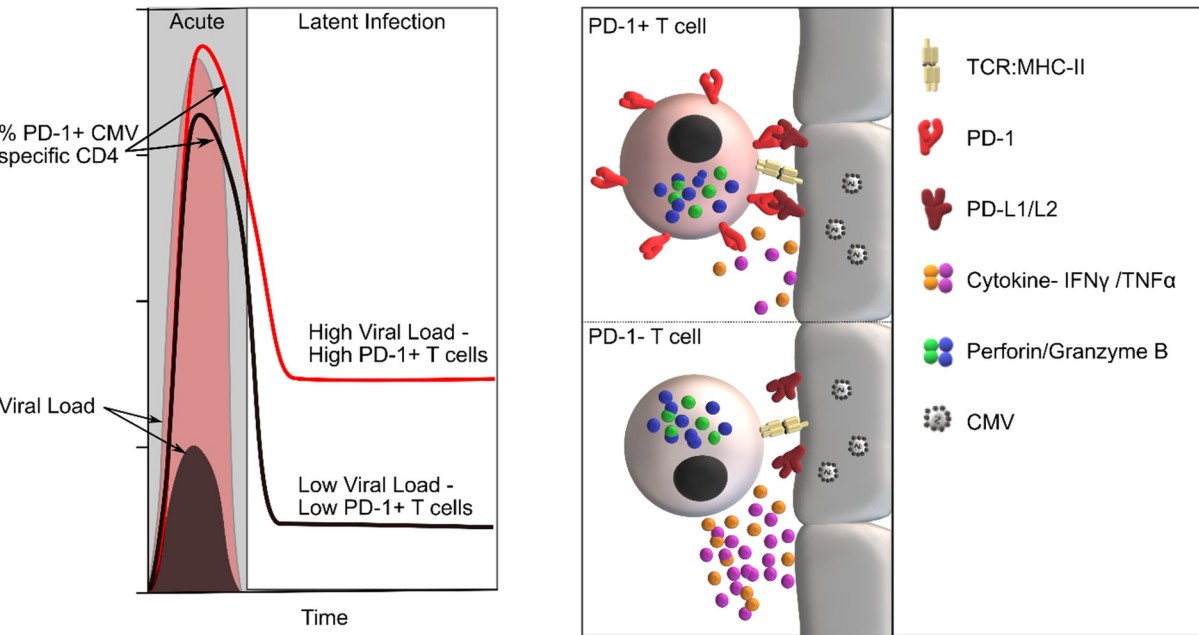

**Fig 7. Model of the association of viral load with the PD-1+ population, and functional relevance.** (a) Viral load during primary infection defines the PD-1 'set point' of the subsequent CMV-specific CD4+ T cell response, high viral load results in higher PD-1+ CD4+ T cell populations. (b) CMV specific cells encountering target cells. PD-1+ T cells possess equal cytotoxic, but display lower release of cytokine to the microenvironment following interaction with PD-1 ligand expressing target cells.

investigation. Of note, these cells provide an opportunity to study proximal aspects of TCR signaling of peptide-specific cells in relation to PD- engagement.

PD-1 blockade was able to partially restore cytokine production [22,40] but this did not approach the levels produced from PD-1- subpopulations and points towards some potential limitations of clinical PD-1 checkpoint blockade in a physiological rather than exhausted setting. It is interesting to speculate on the potential impact that antibody-mediated PD-1 blockade would have on the immune control of CMV. As indicated above, CMV is rarely a clinical concern in immune competent people but such an approach could potentially be used in immune suppressed patients with uncontrolled CMV viremia. Checkpoint inhibition combined with anti-viral medication has shown to be particularly effective in SIV infection models in rhesus macaques [41] and it may be that a similar combination therapy would be valuable for herpesvirus infection.

It is likely that over 5 billion people harbor a persistent CMV infection at the current time and these findings reveal several original features regarding the establishment, maintenance and physiological function of PD-1 expression on CD4+ T cells in the human immune system, as summarized in Fig 7. These observations will help to guide and interpret therapeutic approaches to reinvigorate functional activity within PD-1+ CD4+ T cells in the clinical setting.

## Materials and methods

### Ethics statement

Written informed consent was obtained prior to enrollment from healthy donors (West Midlands research ethics committee 14/WM/1254) and patients undergoing stem cell transplantation (West Midlands research ethics committee 15/WM/0194).

## Participant samples

Healthy donor participants, median age 71 years, were enrolled. A maximum of 90 ml of blood was taken and samples were prepared using Ficoll-Paque gradient centrifugation (PAA, UK). The plasma fraction was stored at -20˚C for CMV ELISA testing, whilst the peripheral blood mononuclear cells (PBMCs) were cryopreserved in liquid nitrogen. All experiments were performed on frozen samples, except RNA sequencing and the PD-1 blockade assay, for which fresh cells were obtained. T cell clones were generated from donor specific PD-1 high and low CMV-specific CD4+ T cells by limited dilution cloning as previously described [42]. These were then grown in T cell media (RPMI-1640 supplemented with 10% fetal calf serum, 1% human serum, 50U/ml IL-2, 30% MLA and 100IU/ml Penicillin and 100μg/ml Streptomycin).

Clones were generated and re-stimulated as required by addition of $1x10^5$ peptide pulsed LCL and $1x10^6$ irradiated PHA activated PBMC. Lymphoblastoid cell lines (LCL) were generated as previously described by transforming B cells derived from donors of known HLA type (13).

LCL were incubated for 60 minutes at 37˚C with the appropriate peptide at 1μg/ml. Cells were then washed twice and resuspended in T cell media.

For experiments studying the retention of PD-1 by T cell clones, re-stimulation (as above) was defined as day 0, no further stimulation was provided. Clones were fed twice weekly by replacement of half culture volume with T cell media.

For studies to determine the relationship between the magnitude of viremia at primary infection and subsequent PD-1 phenotype, blood was obtained from patients undergoing stem cell transplantation. Patients were recruited if they were CMV seropositive at the time of transplantation and had received a stem cell graft from a CMV seronegative donor. CMV reactivation was diagnosed clinically and viral loads obtained from clinical data. Blood samples were taken 18–26 weeks after resolution of viremia for assessment of PD-1 expression.

## CMV ELISA and HLA typing

As described by Kilgour *et al*, CMV IgG ELISA testing (University of Birmingham) was used to ascertain participant's CMV status [43]. For HLA typing, DNA was extracted from PBMC from donors with confirmed positive CMV serology (GenElute Mammalian Genomic DNA miniprep kit; Sigma-Aldrich, St. Louis, MO USA). DNA was used for class II HLA typing for DR7, DR52b, and DQ6 as previously described [44]. Amplified products were run on an agarose gel and imaged using the Kodak EDAS DC290 for Mac OS9.2.

## CD4 CMV specific T cell immunophenotyping

For identification of epitope specific populations PBMCs were defrosted, washed in PBS and 1 x $10^6$ PBMCs were then stained with Live-dead fixable cell dye (Invitrogen). Cells were washed in PBS and re-suspended in 50μL of human serum, together with appropriate tetramer. Cells were then incubated at 37˚C in the dark for 1hr. After repeat washing, surface antibodies were added (as detailed in Table 1). The following tetramer reagents were obtained through the NIH Tetramer Core Facility and consisted of epitopes derived from CMV glycoprotein B (gB); DYSNTHSTRYV (restricted through DRB1*0701 (DR7)), pp65; AGILARNLVPMVATV (restricted through DRB3*0202 (DR52b)) and LLQTGIHVRVSQPSL (restricted through DQB1*06:02 (DQ6)).

Detection of cytokine was achieved by intracellular staining; cells were fixed and permeabilised with 4% PFA and 0.5% saponin, prior to addition of appropriate antibodies. For transcription factor staining cells were fixed and permeabilised with a FoxP3/transcription factor staining buffer set (eBioscience) according to manufacturer's instructions. All flow

**Table 1. Flow cytometry.**

| Antigen | Fluorophore | Clone | Supplier |
|---|---|---|---|
| CCR-7 | Fitc | FAB197F | R&D systems |
| CD14 | Pacific Blue APC-Cy7 | HCD14 | Biolegend |
| CD160 | AF647 | BY55 | Biolegend |
| CD19 | Pacific Blue APC-Cy7 | HIB19 | Biolegend |
| CD244 | FITC | C1.7 | Biolegend |
| CD27 | APC | O323 | Biolegend |
| CD28 | ECD | CD28.2 | Beckman Coulter |
| CD3 | AmCyan | SK7 | BD Biosciences |
| CD3 | FITC BV605 | UCHT1 | Biolegend |
| CD38 | PE-Cy7 PerCP-Cy5.5 | HIT2 | Biolegend |
| CD4 | PE-CF954 | SK3 | BD Biosciences |
| CD4 | AF700 APC-Cy7 Percp-cy5.5 | SK3 | Biolegend |
| CD45RA | AF700 | HI100 | Biolegend |
| CD45RA | APC-Cy7 | HI100 | Biolegend |
| CD45RO | Qdot 605 | UCHL1 | Biolegend |
| CD56 | Pe-Cy7 | HCD56 | Biolegend |
| CD69 | AF647 | FN50 | Biolegend |
| CTLA4 | PE-Cy7 | BN13 | Biolegend |
| Eomes | FITC | WD1928 | eBioscience |
| Granzyme B | AF647 | GB11 | Biolegend |
| Hobit/ZNF683 | AF647 | Sanquin-Hobit/1 | BD Bioscience |
| IFNγ | FITC | B27 | Biolegend |
| IL-2 | PE-Cy7 | MQ1-17H12 | Biolegend |
| LAG3 | PerCP-Cy5.5 | eBioC9B7W | eBioscience |
| PD-1 | PerCP-Cy5.5 PE-Cy7 BV421 | EH12.2H7 | Biolegend |
| Perforin | FITC | dG9 | Biolegend |
| T-bet | Percp/Cy5.5 | 4B10 | Biolegend |
| Tim-3 | APC | F38-2E2 | eBioscience |
| TNFα | APC | MAb11 | eBioscience |

experiments were performed on the LSR II and analysed using BD FACSDiva software (BD Biosciences).

## Peptide stimulation

LCLs were incubated for 30 minutes at 37°C with the appropriate peptide at 1μg/ml. Cells were then washed twice and resuspended in RPMI with $1\times10^6$ donor PBMC cells at a 1:5 ratio and incubated at 37°C for 4 hours. After 1 hour of the incubation, brefeldin A was added. Cells were subsequently washed and stained before analysis by flow cytometry.

## Ex vivo cytotoxicity assay by CMV-specific CD4+ T cells

PD1+ and PD1- tetramer-positive CMV specific CD4+ T cell cells were sorted using a FACS-Melody sorter (BD bioscience). Sorted T cells were co-cultured over night with CFSE

(Invitrogen) labelled peptide pulsed LCL in triplicate. CFSE-labelled LCL without peptide were used as control. Live CFSE-labelled LCLs were determined by propidium iodide (PI) staining and quantified using a BD Accuri flow cytometer (BD Biosciences). Specific lysis was calculated according to the count of the control culture by $100 \times (1 - $(experimental group cell count/control cell count)).

### *Ex vivo* PD-1 blockade assays

Using fresh PBMCs, cells were incubated overnight with CMV lysate (in house) with or without 20 μg/mL of anti-PD-1 (Biolegend). Cells were then washed and stained with the same surface antibodies for the peptide stimulation assay, before fixation and permeabilisation with PFA and saponin. Cells were then stained for IFN-γ and analysed by flow cytometry. For the PD-1 blocking experiments using CMV-specific T cell clones, DYS specific CD4 T cells were stimulated with DYSN peptide pulsed LCLs before the concentrations of IFN-γ in the culture were compared before and after PD-1 blockade using ELISA.

### RNA sequencing

PD1+ and PD1- CMV-specific CD4+ T cell responses from 6 donors were obtained for analysis. Using freshly isolated PBMCS, cells were firstly enriched for CD4+ T cells using the Easysep human CD4+ enrichment kit (Stemcell technologies, Cambridge UK). Enriched CD4+ T cells were then stained with CMV Class II tetramer, and subsequently surface antibodies for 15 minutes at 4˚C. Using the MoFlo sorter, CD3+, CD4+, tetramer+ cells were sorted into 2 populations; PD1+ and PD1-. Sorting was performed within 4 hours of blood sampling. After sorting, cells were immediately pelleted, 3500g for 10 minutes, and RLT plus lysis buffer added and vortexed for 1 minute. RNA extraction was then performed according to manufacturer's instructions. (RNEasy Plus micro kit, Qiagen) RNA retrieved was amplified and sequenced on the Illumina HiSeq2000 platform using TruSeq version 3 chemistry, over 100 cycles (Oxford Gene Technologies). Read quality was assessed with FastQC [45] and Prinseq [46] was used to trim reads with low quality bases. The alignment of reads to the reference genome hg19 was carried out using STAR [47] and samples with low % of reads mapping to the reference genome were discarded from the analysis (1 donor). Raw counts of reads aligning to gene features were calculated using HTSeq [48] and differential gene expression analysis was performed using the R package DESeq2 [49]. Functional enrichment analysis of differentially expressed genes (p<0.001), was performed using g:Profiler [50]. For gene set enrichment analysis, the Bioconductor R package, GAGE [51] was used to identify coordinated enrichments of published gene sets retrieved from the Molecular Signatures Database [52].

### T cell receptor sequencing

Class-II tetramer+ cells were sorted on a BD FACSMelody cell sorter into 96 well PCR plates containing 10ul/well of a 10mM Tris and 500U/ml RNasin Inhibitor solution. Cells were sorted to give equal number of PD-1+ or PD-1- cells from each donor. PD-1 levels was determined as the highest and lowest 20–30% of the population. Plates were then spun at 340g for 5mins and stored at -80˚C prior to cDNA synthesis.

TCR sequencing was performed as previously described [53]. Briefly, nested PCR of the TCR alpha and beta CDR3 region was performed by firstly using a Onestep RT-PCR kit (Qiagen) for cDNA synthesis and first round PCR, followed by second amplification with HotStar Taq (Qiagen). Finally a barcoding reaction was carried out using HotStar Taq. PCR products were then pooled and gel purified (Qiagen) prior to running a paired end sequencing reaction using a MiSeq v2 500 sequencing kit (Illumina). Paired-end TCR sequencing reads were

grouped by experiment into forward and reverse pairs, demultiplexed according to their single cell specific barcodes and assigned to the Plate-Well-single cell source using a bespoke python script. These reads were aligned to germline segments of human TCR sequences and clonotypes assembled with MiXCR [54]. MiXCR output was imported into R (v3.5.1) [55] to tally the number of clonotypes identified and graphical outputs were generated with ggplot2 [56]. Circos plots were generated with the circlize package (v0.4.8) [57].

## Supporting information

**S1 Fig. Validation of Tetramer-specific binding.** Non-specific tetramer binding was assessed by co-staining with an irrelevant tetramer. (A) Gating strategy used to determine non-specific binding. Tetramer-positive cells were gated (left) and the binding of irrelevant tetramer (Irr Tet) assessed in this population (right). (B) An example of staining of tetramer and irrelevant tetramer in bulk CD4 T cells. (C) Shows summary data for individual tetramers (DYS n = 4, AGI n = 3, LLQ n = 4). (D) The percentage of PD-1+ cells was assessed in the total population of tetramer-binding cells and also in the 'specific' population from which the population of irrelevant-tetramer binding cells had been removed. No significant difference was observed in relation to the percentage of PD-1 expression on these two populations (paired t-test).
(TIF)

**S2 Fig. PD-1 status is not associated with Memory phenotype.** The memory status of PD-1 + and PD-1- CMV Tetramer+ CD4+ T cell subsets was determined on healthy donors (n = 21) by expression of CD45ra and CCR7; Naïve—CD45ra+CCR7+, Tcm -CD45ra-CCR7+, Tem —-CD45ra-CCR7-, Temra—CD45ra+CCR7-.
(TIF)

**S3 Fig. Inhibitory receptor expression by CMV-specific CD4+ T cells.** Example plots of the expression of inhibitory receptors on CMV tetramer-specific CD4+ T cells.
(TIF)

**S4 Fig. Frequency of tetramer-positive CD4+ T cells and relative CD28 expression are not related to peak viral load or percentage PD-1 expression.** Correlation of parameters from CMV acute infection in bone marrow transplant patients (n = 5) at 25 weeks post resolution of viremia. (**A**) Peak CMV viral load and frequency of Tetramer+ CD4 T cells. (**B**) Frequency of PD-1+ Tetramer+ CD4 T cells and Tetramer+ CD4 T cell frequency. (C) Frequency of PD-1 + Tetramer- CD4 T cells and Peak CMV viral load. (D) CD28+ Tetramer+ cells and Tetramer + CD4 T cell frequency. No correlation was observed.
(TIF)

**S5 Fig. PD-1 Ligand expression on PBMC subsets.** Expression of the PD-1 ligands PD-L1 and PD-L2 was assessed on monocyte (CD14+CD19-CD3-), B cell (CD19+CD14-CD3-) and T cell (CD3+ CD14-CD19-) populations. Expression was found consistently on monocyte and B cell populations. Shown are example plots and combined data from five healthy donors.
(TIF)

**S6 Fig. Gating strategy for sorting of PD-1+ and PD-1- tetramer+ subsets.** This was used for cell sorting for T cell cloning, single cell TCR sequencing and RNA-seq experiments.
(TIF)

**S7 Fig. PD-1+ and PD-1- CMV Tetramer+ CD4+ T cell clones show similar clonality to *ex vivo* sorted T cells.** PD-1+ and PD-1- DYS and AGI specific CD4+ T cells cloned by limited dilution *ex vivo* were single cell sorted. Shown are *TCRA* and *TCRB* sequencing of individual T

cell clones. No clonotypes are shared between AGI PD-1+ and PD-1- T cell clones, whereas DYS clones show shared TCR usage between PD-1+ and PD-1- T cell clones.
(TIF)

**S8 Fig. Marker validation and Gene Set Enrichment Analysis (GSEA) of selected hallmark, pathway and immune signature gene sets from the Molecular Signatures Database. (**A) Flow cytometry was used to assess protein expression of a selection of differentially regulated genes on PD-1+ and PD-1- CMV tetramer-positive CD4+ T cells. P values were calculated by two tailed paired t test, * p = <0.05 ** p = <0.005. (B) Selected gene sets were analysed for their enrichments within ranked genes from differential expression analysis between PD1+ vs PD1- CMV tetramer+ CD4+ T cells. Points indicate the log2 fold change (PD1+/PD1-) in expression of genes within the gene set. GSEA false discovery rate (FDR) is reported for gene sets with FDR < 0.3 indicating their coordinated upregulation (↑) or downregulation (↓) in PD1+ cells.
(TIF)

# Author Contributions

**Conceptualization:** Helen M. Parry, Alexander C. Dowell, Jianmin Zuo, Paul Moss.

**Data curation:** Helen M. Parry, Alexander C. Dowell, Jianmin Zuo, Wayne Croft, Archana Sharma-Oates.

**Formal analysis:** Helen M. Parry, Alexander C. Dowell, Jianmin Zuo, Kriti Verma, Francesca A. M. Kinsella, Jusnara Begum, Wayne Croft, Archana Sharma-Oates.

**Funding acquisition:** Helen M. Parry, Guy Pratt, Paul Moss.

**Investigation:** Helen M. Parry, Alexander C. Dowell, Jianmin Zuo, Kriti Verma, Francesca A. M. Kinsella, Jusnara Begum, Wayne Croft.

**Resources:** Guy Pratt, Paul Moss.

**Supervision:** Guy Pratt, Paul Moss.

**Visualization:** Helen M. Parry, Alexander C. Dowell.

**Writing – original draft:** Helen M. Parry, Alexander C. Dowell, Paul Moss.

**Writing – review & editing:** Helen M. Parry, Alexander C. Dowell, Jianmin Zuo, Kriti Verma, Francesca A. M. Kinsella, Jusnara Begum, Wayne Croft, Paul Moss.

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
