## [Decision Letter · Decision Letter 0]

12 Jun 2020

Dear Dr Dowell,

Thank you very much for submitting your manuscript "PD-1 is imprinted on cytomegalovirus-specific CD4+ T cells and attenuates Th1 cytokine production whilst maintaining cytotoxicity" for consideration at PLOS Pathogens. As with all papers reviewed by the journal, your manuscript was reviewed by members of the editorial board and by several independent reviewers. In light of the reviews (below this email), we would like to invite the resubmission of a significantly-revised version that takes into account the reviewers' comments.

Three reviewers have independently assessed your manuscript. They concur that your observations are interesting, but that several issues require additional experimental evidence to strengthen the conclusions. In particular you are encouraged to generate direct evidence that demonstrates the functionality of PD-1 expressing CD4 T cells. In light of that, I kindly request that you address all comments and provide additional experimental evidence where necessary.

We cannot make any decision about publication until we have seen the revised manuscript and your response to the reviewers' comments. Your revised manuscript is also likely to be sent to reviewers for further evaluation.

Sincerely,

Luka Cicin-Sain, M.D:, Ph.D:

Guest Editor

PLOS Pathogens

Klaus Früh

Section Editor

PLOS Pathogens

Kasturi Haldar

Editor-in-Chief

PLOS Pathogens

orcid.org/0000-0001-5065-158X

Michael Malim

Editor-in-Chief

PLOS Pathogens

orcid.org/0000-0002-7699-2064

Three reviewers have independently assessed your manuscript. They concur that your observations are interesting, but that several issues require additional experimental evidence to strengthen the concldusions. In particular you are encouraged to generate direct evidence that would demonstrate the functionality of PD-1 expressing CD4 T cells. In light of that, I kindly request that you address all comments and provide additional experimental evidence where necessary.

Reviewer's Responses to Questions

**Part I - Summary**

Reviewer #1: This paper from Moss and colleagues uses HLA-II tetramers recently generated by their group to characterize the PD-1 expression by HCMV epitope-specific CD4 T cells isolated from peripheral blood of normal healthy donors, or persons undergoing a primary CMV infection upon transplant. They report that PD-1 is expressed on a subset of these cells, that the initial viral load in viremic transplant patients likely regulates the setpoint of PD-1 expression which then remains stable over time, that PD-1+ cells produce less effector cytokines than their negative counterparts when restimulated ex vivo, that this diverse PD-1 expression may or may not depend upon initial CD4 T cell priming based on TCR sequencing/clonotyping, and that these PD-1 subsets show distinct RNA transcriptomes. This work advances our understanding of how PD-1 expression by HCMV CD4 T cells correlates with CD4 T cell phenotype, and with some additional experimentation showing how these phenotypes may be tied directly (or not) to PD-1 mediated signaling could be a significant advance for the field.

Reviewer #2: Parry and colleagues studied the role of PD-1 on cytomegalovirus (CMV)-specific T cells using class II tetramers to identify the cells and combine this with cell surface marker expression. Moreover, phenotypical and functional assays were performed and showed that the PD-1+ CD4 T cells are equally cytotoxic but produce lower levels of IFN-g/TNF as compared to PD-1- CD4 T cells. Thus far, little is known about PD-1+ CD4 T cells during CMV infection.

Reviewer #3: In the manuscript by Parry et al the expression of the inhibitory receptor PD-1 on Human Cytomegalovirus specific CD4+ T cells (identified by MHC Class II peptide loaded tetramers) is investigated. The manuscript considers expression following primary infection, the effect on virus load, the stability of expression over time, TcR usage in PD-1 +/- antigen specific CD4 T cells as well as effector functions and gene expression profiles by RNA-seq.

The work presented is an observational study – key findings are that there is a stable set point for proportions and antigen specific cells that express PD-1 that depending on antigen specificity the same TcR clonotypes clonotype can be represented in PD-1 +/- subsets and that these cells do not express many hallmarks associated with “exhaustion”.

A general point which should be made clear in the initial results is that the tetramers available for this work are very limited in terms of the HCMV proteins that are recognized by CD4+ T cells, 1 from gB and 2 from pp65. As such some caution should be exercised in the discussions and conclusions drawn more generally.

However, in general the manuscript is well written, the data is clearly presented, and the analysis and methods used appropriate. The overall conclusions are supported with some caveats required.

**Part II – Major Issues: Key Experiments Required for Acceptance**

Reviewer #1: -The authors raise the hypothesis in the discussion and in Fig7 that perhaps PD-1+ CD4 T cells evolve to kill/clear CMV infected cells in the absence of robust inflammatory cytokine expression (e.g. IFNg), in particular they point to PD-L1+ endothelium and maintenance of vascular integrity. This is an intriguing idea, but if correct may then require a role for PD-L1/PD-1 signaling in the modulation of CTL activity or suppression of cellular signaling molecules in these CD4T cells. The authors saw no difference in cytotoxic molecules or direct killing of peptide-pulsed targets for PD-1+ or (-) cells in Fig 3DE. This peptide killing assay needs to be performed including a PD-1 agonist antibody to see if this enhances the CTL activity of the PD-1+ subset. Additionally, the activation/phosphorylation of key STAT or other TCR proximal adaptors needs to be examined in the PD-1+ and (-) cells where IFNg and TNF were only assessed in the presence of a PD-1 agonist mAb. Additionally, the authors need to test whether autologous APC infected with HCMV are more or less sensitive to killing by PD-1+ or (-) subsets, or produce different effector cytokines, as this is the real test, not the ability to kill uninfected peptide pulsed cells. There are many intriguing observations in this paper, but more direct evidence that these PD-1+ and (-) expressing CD4 T subsets mediate distinct effector functions is needed to supports the physiological importance.

-Although there is significant evidence from many groups that CMV CD4 T cells can be cytolytic, many in the field still debate the importance/relevance of this as compared to classic CD8 CTL. Therefore, it would be extremely valuable if the authors could show how the absolute expression levels of perforin and gzm reported for CD4 tet+ cells in Fig 3D compare to a canonical HCMV-specific CD8 T cell from the same donor (e.g. pp65 or IE1 specific). Going a step further and comparing their relative per-cell killing potential compared to a HCMV CD8 T would be great in this regard as well. Even if the authors find these HCMV CD4T are not as ‘good’ as CD8 in their killing assay, or don’t express as high levels of classical cytotoxic effector molecules, this would not diminish the importance of their results in this reviewers mind. It would simply give the field a relative feeling for how to think/compare these two importance T cell effector subsets in the context of CMV infection.

-When analyzing HCMV-specific CD4 T cell using HLA-II tetramers, the authors only do a ‘single-staining’ analysis, using the tetramer labeled with only a single fluorophore (as shown in Fig 1A). As has been shown by the Jenkins group and others, this results in the detection of a significant number of false-positive cells which bind the tetramer non-specifically. Therefore, it is becoming the near-standard now to label the same tetramer with two separate fluorphores, and also include a 3rd non-specific tetramer (e.g. CLIP) labeled with a 3rd fluorphore, and then only identify cells that are double+ for the HCMV tetramer and negative for the CLIP as HCMV epitope-specific CD4 T cells. As most all of the authors conclusions are based on this tetramer staining analyses, it is critical that they repeat at least a few of their experiments using this 3-tetramer staining method to verify that the proportions and phenotypes of PD-1+ and PD-1(-) cells don’t change significantly. In addition, again in Fig 1A, there is no clear distinction between PD-1+ and (-) cells in the total CD4 T population. Certainly there is no clear PD-1 hi vs lo populations as you see in exhausted CD4 or CD8 T cells in tumor studies (as noted in their discussion). A bit more clarity is needed on how the authors ascribe a cell being PD-1+ or (-) to define their percentages, as no gate is shown in 1A? For instance, showing how they gated on + vs (-) cells for their RNAseq experiments.

Reviewer #2: - In 1C the proportion of Tet+ CD4 T cells expressing one or more inhibitory receptors are plotted in bar graphs but it is complete unclear what the percentages of the individual inhibitory receptors are.

-In 2A a correlation of PD-1 on the CMV-specific CD4 T cells with the peak viral load is suggested based on 5 samples. The data is interesting but more data should be added.

-What is the correlation between PD-1 on the CMV-specific CD4 T cells and the percentage of the CMV-specific CD4 T cells. Likewise what is the correlation between the peak viral load and the percentage of the CMV-specific CD4 T cells.

-2E/Fig3. CD28- are considered as cytotoxic CD4+ T cells and the PD-1+ T cells have lower expression of CD28. Still, a comparable cytotoxicity is shown in Fig 3. Do the authors have an explanation for this?

-In the same line and based on the possible like between PD-1 and CD28 expression; is there a correlation between CD28 expression on the CMV-specific CD4 T cells with peak viral load and/or percentage of CMV-specific CD4 T cells.

-In Figure 4 it is shown that the PD-1+ CD4 T cells produce lower amounts of IFN-g and TNF. Is this also found for IL-2? The latter is of interest since IL-2 is known as the most prominent cytokine to be negatively influenced by PD-1.

-The RNA sequencing data in fig 6 indicates that certain transcripts are lower in PD-1+ CD4 T cells while other are higher. The authors wrote that 10 genes were upregulated in PD-1+ cells and 5 genes were downregulated in PD-1+ cells. However, the volcano plot show that 4 genes were downregulated and 11 genes upregulated. Importantly, there is no validation of these results (except for CD28 shown in Fig 2). RT-PCR and/or protein expression studies could be done to validate at least more than 1 hit.

Reviewer #3: Figure 2

(A) I think that the authors need to interpret this result with much more caution. 3 patients have very low peak loads 2 of which have over 50% of tetramer cells PD-1 + , while 1 patients has a peak 60.10e3 but only a slightly higher % of PD-1 + cells and the other is indeed higher again. In addition these are immunosuppressed transplant patients and indeed the authors do make the point that PD-1 expression is affected in this group later on in the results section. The authors should discuss this and that data from primary infection (I appreciate this is hard) but maybe kidney transplant primary infection which has a much lower immune suppression regime could help support this hypothesis.

Figure 4 attenuation of cytokine production in PD-1 = T cells.

LCL cells lines which are EBV transformed have been used as antigen presenting cells (as they express PD-L1) in these experiments pulsed with CMV peptides and co culture with whole PBMC. The readout is intracellular IFN gamma or TNFa – however this would be measuring T cell responses to not only the HCMV peptides but also to EBV presented antigens – as such the results between PD-1 +/- are a mixture of T cells specific to 2 different herpesviruses. However, in the figure legend it states that these were HCMV tetramer positive cells, which would be a much more logical experiment. Could the authors please clarify the experiment and have the results and figure legend match.

In the blocking experiments CMV lysate is used to pulse whole PBMC given the previous experiment used LCLs as they had PD-L1 expression, what is the expression of PD-L1 on subsets in PBMC. The effect of blocking antibody in this experiment shows good results in only 3 of the donors tested while 4 others have minimal increases in IFN g? The authors should address these points.

**Part III – Minor Issues: Editorial and Data Presentation Modifications**

Reviewer #1: -Can the authors please describe the method for how the limit-dilution cloned PD-1+ and (-) cells that were FACS sorted were kept alive for 4 weeks to assess the maintenance of PD-1 expression. Were they restimulated weekly with autologous APC+peptide (LCL in methods?)…IL-2..something else?

-The authors show in Fig 1D that a small % of tet+ HCMV CD4T also express one or more other markers which are often used to identify ‘exhausted’ T cells. The raw FACS histograms where they gated/identified these cells as + for these other markers needs to be shown, because small increases in MFI would not be consistent with the conclusion that these cells are more exhausted, as this is a similar case to PD-1+ vs +++ for instance.

Reviewer #2: The authors should better indicate which groups are plotted in 1B (e.g. Tet+ and Tet- could be used instead of Tet and CD4).

Reviewer #3: End of first results section “…CD4+ T cells is not related to activation status or exhaustion.” Could you amend this to “…CD4+ T cells is not related to markers of activation status or markers of exhaustion.”

PLOS authors have the option to publish the peer review history of their article (what does this mean?). If published, this will include your full peer review and any attached files.

Reviewer #1: No

Reviewer #2: No

Reviewer #3: No
---

## [Decision Letter · Decision Letter 1]

2 Nov 2020

Dear Dr Dowell,

Thank you very much for submitting your manuscript "PD-1 is imprinted on cytomegalovirus-specific CD4+ T cells and attenuates Th1 cytokine production whilst maintaining cytotoxicity" for consideration at PLOS Pathogens. As with all papers reviewed by the journal, your manuscript was reviewed by members of the editorial board and by several independent reviewers. The reviewers appreciated the attention to an important topic. Based on the reviews, we are likely to accept this manuscript for publication, providing that you modify the manuscript according to the review recommendations.

**Two reviewers have expressed satisfaction with your revised version. There is only a relatively minor point brought up by one of the reviewers that is still outstanding. Please address this issue in a revised version.**

Sincerely,

Luka Cicin-Sain, M.D:, Ph.D:

Guest Editor

PLOS Pathogens

Klaus Früh

Section Editor

PLOS Pathogens

Kasturi Haldar

Editor-in-Chief

PLOS Pathogens

orcid.org/0000-0001-5065-158X

Michael Malim

Editor-in-Chief

PLOS Pathogens

orcid.org/0000-0002-7699-2064

Dear Authors,

The reviewers have expressed satisfaction with your revised version. There is only a relatively minor point brought up by one of the reviewers.

Please address this issue in a revised version.

Sincerely

Luka Cicin-Sain

Reviewer Comments (if any, and for reference):

Reviewer's Responses to Questions

**Part I - Summary**

Reviewer #1: The authors have done a largely good job addressing my initial concerns. I only ask that they provide evidence that their experiments using the anti-PD1 blocking antibody actually neutralized signaling in their system, as the experiment I had asked for was to include an anti-PD1 agonist, not to block PD-1 as they have in their new experiments presented in Fig 3F.

Reviewer #2: The authors have addressed all my comments/suggestions.

Reviewer #3: The authors have addressed each of my comments providing additional explanation and changes to the manuscript text in addition to performing the additional experiment suggested - which did indeed demonstrate PD-1L expression in B and Monocytes in PBMC.

I am satisfied with the changes made to the manuscript

**Part II – Major Issues: Key Experiments Required for Acceptance**

Reviewer #1: (No Response)

Reviewer #2: (No Response)

Reviewer #3: (No Response)

**Part III – Minor Issues: Editorial and Data Presentation Modifications**

Reviewer #1: (No Response)

Reviewer #2: (No Response)

Reviewer #3: (No Response)

PLOS authors have the option to publish the peer review history of their article (what does this mean?). If published, this will include your full peer review and any attached files.

Reviewer #1: No

Reviewer #2: No

Reviewer #3: No
---

## [Editor Report · Decision Letter 2]

1 Feb 2021

Dear Dr Dowell,

We are pleased to inform you that your manuscript 'PD-1 is imprinted on cytomegalovirus-specific CD4+ T cells and attenuates Th1 cytokine production whilst maintaining cytotoxicity' has been provisionally accepted for publication in PLOS Pathogens.

Best regards,

Luka Cicin-Sain, M.D:, Ph.D:

Guest Editor

PLOS Pathogens

Klaus Früh

Section Editor

PLOS Pathogens

Kasturi Haldar

Editor-in-Chief

PLOS Pathogens

orcid.org/0000-0001-5065-158X

Michael Malim

Editor-in-Chief

PLOS Pathogens

orcid.org/0000-0002-7699-2064
---

## [Editor Report · Acceptance letter]

16 Feb 2021

Dear Dr Dowell,

We are delighted to inform you that your manuscript, "PD-1 is imprinted on cytomegalovirus-specific CD4+ T cells and attenuates Th1 cytokine production whilst maintaining cytotoxicity," has been formally accepted for publication in PLOS Pathogens.

Best regards,

Kasturi Haldar

Editor-in-Chief

PLOS Pathogens

orcid.org/0000-0001-5065-158X

Michael Malim

Editor-in-Chief

PLOS Pathogens

orcid.org/0000-0002-7699-2064